METHODS AND RESOURCES

# Single-cell analysis of isoform switching and transposable element expression during preimplantation embryonic development

**Chaoyang Wang**[1,2,3☯]**, Zhuoxing Shi**[3☯]**, Qingpei Huang**[1,2]**, Rong Liu**[1,2]**, Dan Su**[1,2]**, Lei Chang**[1,2]**, Chuanle Xiao**[3]**, Xiaoying Fan** [1,2,4]*

**1** GMU-GIBH Joint School of Life Sciences, The Fifth Affiliated Hospital of Guangzhou Medical University, Guangzhou Laboratory, Guangzhou Medical University, Guangzhou, China, **2** The Bioland Laboratory (GuangZhou Regenerative Medicine and Health Guangdong Laboratory), Guangzhou, China, **3** State Key Laboratory of Ophthalmology, Zhongshan Ophthalmic Center, Sun Yat-sen University, Guangdong Provincial Key Laboratory of Ophthalmology and Visual Science, Guangzhou, China, **4** The Guangzhou Institutes of Biomedicine and Health, Chinese Academy of Sciences, Guangzhou, China

☯ These authors contributed equally to this work.
* fan_xiaoying@gzlab.ac.cn

**Data Availability Statement:** All single-cell isoform sequencing data generated in this study are available at NCBI Gene Expression Omnibus (https://www.ncbi.nlm.nih.gov/geo/) under the

## Abstract

Alternative splicing is an essential regulatory mechanism for development and pathogenesis. Through alternative splicing one gene can encode multiple isoforms and be translated into proteins with different functions. Therefore, this diversity is an important dimension to understand the molecular mechanism governing embryo development. Isoform expression in preimplantation embryos has been extensively investigated, leading to the discovery of new isoforms. However, the dynamics of isoform switching of different types of transcripts throughout the development remains unexplored. Here, using single-cell direct isoform sequencing in over 100 single blastomeres from the mouse oocyte to blastocyst stage, we quantified isoform expression and found that 3-prime partial transcripts lacking stop codons are highly accumulated in oocytes and zygotes. These transcripts are not transcription by-products and might play a role in maternal to zygote transition (MZT) process. Long-read sequencing also enabled us to determine the expression of transposable elements (TEs) at specific loci. In this way, we identified 3,894 TE loci that exhibited dynamic changes along the preimplantation development, likely regulating the expression of adjacent genes. Our work provides novel insights into the transcriptional regulation of early embryo development.

## Introduction

A gene can be transcribed into various isoforms, which are then translated into different proteins. Isoform compositions differ between cell types and states, making isoform switching a crucial factor in determining cell identity [1,2]. Third-generation sequencing-based single-cell RNA-sequencing methods like SCAN-seq, HIT-scISOseq, and MAS-ISO-seq have been developed to directly sequence gene isoforms [1,3–7]. SCAN-seq is known for its high gene

accession number GSE250381. Code availability The HIT-scISOseq analysis pipeline and source code are available from https://github.com/shizhuoxing/scISA-Tools. Additionally, the source code utilized in this study has been published at: https://zenodo.org/records/10394889.

**Funding:** This work was supported by grants from the National Key Research and Development Program of China (2020YFA0112201 to XF), the National Natural Science Foundation of China (32071451 to XF), the Guangdong Provincial Pearl River Talents Program (2021QN02Y747 to XF), and the R&D Program of Guangzhou National Laboratory (SRPG21-001 to XF). The funders had no role in study design, data collection and analysis, decision to publish, or preparation of the manuscript.

**Competing interests:** The authors have declared that no competing interests exist.

**Abbreviations:** CCS, circular consensus sequencing; E2C, early 2-cell; GO, Gene Ontology; hCG, human chorionic gonadotropin; L2C, Late 2-cell; LINE, long interspersed element; LTR, long terminal repeat; mESC, mouse embryonic stem cell; MZT, maternal to zygote transition; ORF, open reading frame; PCA, principal component analysis; PMSG, pregnant mare's serum gonadotropin; RRM, RNA recognition motif; SINE, short interspersed element; SRA, Sequence Read Archive; TE, transposable element; TSS, transcription start site; ZGA, zygotic genome activation.

detection sensitivity and ability to detect many novel transcripts in rare samples [4]. However, it fails to quantify the absolute abundance of genes and isoforms due to the high error rate of Nanopore sequencing [8,9]. On the other hand, HIT-scISOseq and MAS-ISO-seq use the PacBio HiFi sequencing platform to quantify isoform abundance in single cells with improved data throughput [6,7].

Isoform switch plays an important role in cell fate determination. *PBX1*, for example, can be transcribed into 3 different isoforms, each with distinct functions. *PBX1a* maintains the pluripotency of mouse embryonic stem cells (mESCs), while *PBX1b* promotes differentiation [10]. Other genes such as *Tcf3* and *Sall4* have similar regulatory patterns in mESCs [11,12]. The molecular regulation of preimplantation embryo development has been the focus of many studies, particularly maternal to zygote transition (MZT), which is crucial for whole-body development [13–16]. Although hundreds of genes have been identified in zygotic genome activation (ZGA), the functional regulators remain largely unclear, including whether isoform switching participates in the process [17–19].

Transposable elements (TEs) account for approximately 46% and approximately 37.5% of the human and mouse genome, respectively [20,21], contributing to evolution and genetic regulation. They can be divided into 2 major classes based on the transposition mode: class I retrotransposon and class II DNA transposon [22,23]. Class I, which makes up about 95% of total TEs, includes long and short interspersed elements (LINEs and SINEs, respectively) and long terminal repeats (LTRs). TEs are known to play a crucial role during embryo development [24]. For instance, MERVL and MT2_mm (a truncated form of ERVL containing only the LTR domain) can serve as promoters for totipotent gene expression, and their expression has been considered an essential totipotent biomarker [18,25,26]. LINEs, particularly LINE1, have been reported to suppress the expression of totipotent genes such as *Dux* [27,28]. A previous study showed that the hominoid-specific transposon (SINE-VNTR-Alu) acts as an enhancer to promote the ZGA process [29]. However, due to their highly repetitive nature, it is challenging to determine the activity of TEs at the locus level with limited read length. Analyzing TE expression in specific loci is therefore important for gene transcriptional regulation.

In this study, we adapted the HIT-scISOseq method for low-throughput cell analysis and sequenced isoforms in single blastomeres of mouse preimplantation embryos [6]. We analyzed cell heterogeneity within the same embryos at the same stage, providing insights into the timing of cell fate diversification during preimplantation embryo development. Isoforms of each gene in every single cell were quantified, and different isoform types showed varied proportions across embryonic stages. Notably, a significant number of 3-prime partial transcripts (lacking stop codons and generating proteins lacking C-termini) were observed in oocytes and zygotes, but were quickly degraded at the early 2-cell (E2C) stage. Furthermore, locus-specific TEs were analyzed, revealing dynamic expression changes during embryonic development. These TEs showed high correlation with the expression of adjacent genes, indicating their potential importance in developmental events.

## Results

### Modified HIT-scISOseq for the mouse preimplantation embryos sequencing

To identify gene isoforms in each blastomere of mouse oocytes and preimplantation embryos, we amplified RNAs in individual cells using a 10× gel bead and the Smart-seq2 protocol [6,30]. Subsequently, the amplified cDNAs from different cells were combined for ligation and PacBio library construction following the HIT-scISOseq method [6]. Concurrently, the barcode sequence of each cell was predetermined through Sanger sequencing of the cDNAs (Fig 1A).

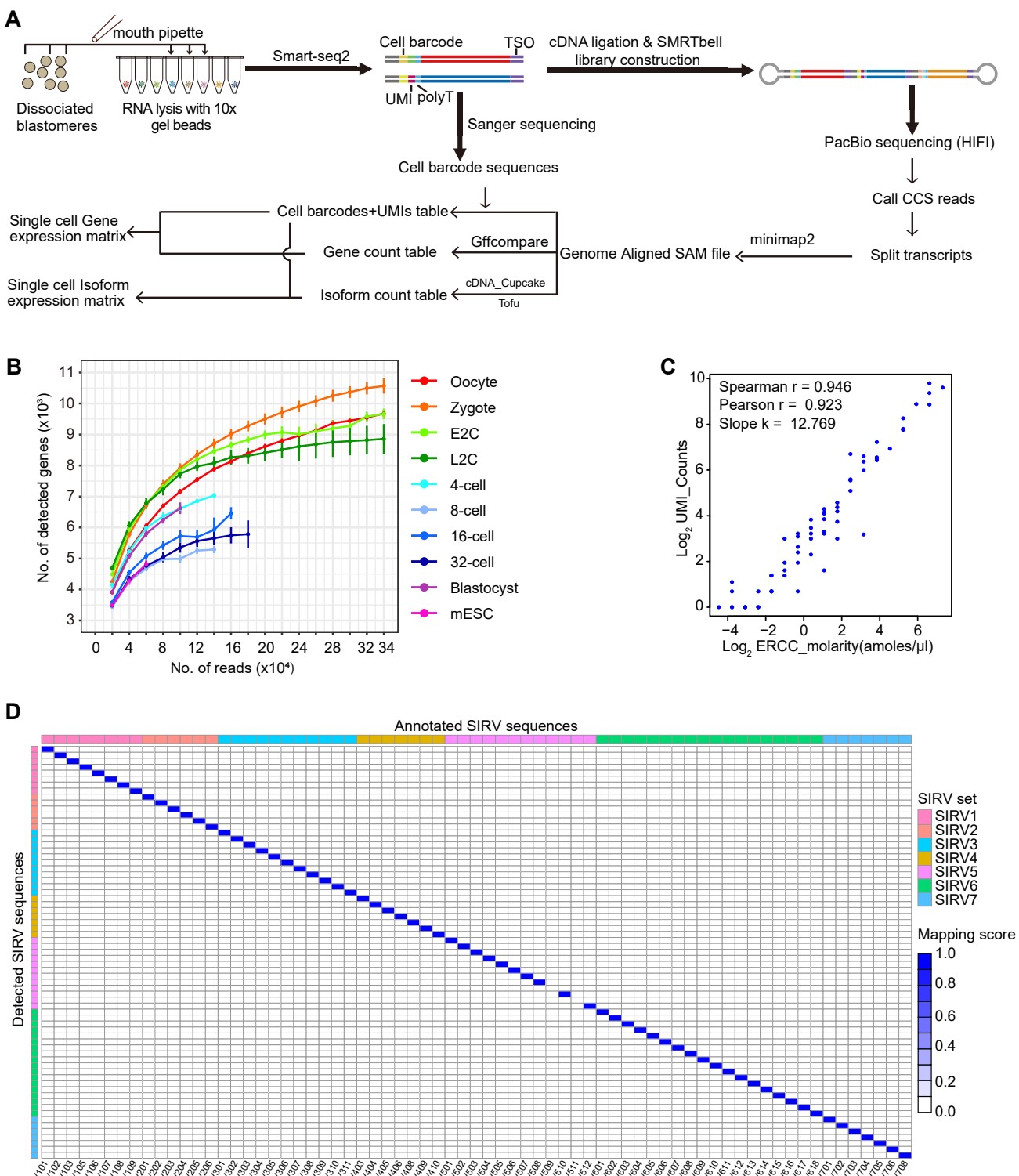

**Fig 1. Quality evaluation of the single-cell isoform expression data.** (A) Diagram of the experimental and analysis workflow for single-cell isoform sequencing of mouse preimplantation embryos. (B) Saturation curve of representative cells from each stage. The raw data for this plot is supplied in S1 Data. (C) Correlation between detected UMI counts and absolute spiked abundances of each ERCC gene. The raw data for this plot is supplied in S1 Data. (D) Isoform mapping results of the SIRV spike-ins. The raw data for this plot is supplied in S1 Data.

Data on gene and isoform expression from 161 single blastomeres were collected from 3 batches, covering various developmental stages and mESC (Table 1). Each sequencing batch produced approximately 5 million circular consensus sequencing (CCS) reads in 1 SMRT Cell 8M, with an average length of around 4 kb, indicating ligation of 2 to 3 cDNA molecules in most cases. After data splitting and mapping, about 90% of the isoforms could be accurately assigned to cells (Table 1). This approach allowed for relatively deep sequencing of samples from each stage (Fig 1B). To assess the precision of measuring absolute numbers of isoforms using our method, we also amplified ERCC and SIRV spike-ins [31,32]. At the gene level, we observed high correlation values between the added molecules and the detected UMI counts (Fig 1C). At the isoform level, different isoforms of the same SIRV gene were accurately identified without any false matches (Fig 1D). These findings demonstrate that our workflow precisely measures transcript abundance in each single cell.

## Gene and isoform expression patterns in the mouse preimplantation embryos

The mouse oocyte and zygote contain a higher number of RNA molecules compared to later stage blastomeres due to maternally inherited RNA degradation [33]. The number of transcript molecules was strongly correlated (R = 0.96) with the number of expressed genes in the cells (Fig 2A). Principal component analysis (PCA) using gene expression data and isoform expression data showed that blastomeres of different stages were clearly separated (Fig 2B and 2C). The oocyte and zygote exhibited similar expression patterns; Late 2-cell (L2C) and 4-cell blastomeres were grouped together; the 8-cell, 16-cell, and 32-cell stages were similar; and the blastocyst cells were analogous to the mESCs. Stage-specific genes and transcripts were extracted, resulting in 3,867 genes and 6,819 isoforms, respectively (S1 Table). These were divided into 6 corresponding clusters based on their expression patterns across all embryonic stages (Fig 2D and 2E and S1 Table). Cluster 1 (C1) transcripts were highly abundant in oocytes and zygotes, subsequently degraded from E2C stage. Cluster 2 (C2) included transcripts that were only up-regulated in the E2C stage. Cluster 3 (C3), cluster 4 (C4), and cluster 5 (C5) transcripts were highly expressed in the L2C to 4-cell stages, 8-cell to 32-cell stages, and blastocyst stages, respectively. The mESC-specific transcripts were in cluster 6 (C6). More genes were identified in each cluster at the isoform level, and most of the isoforms were consistent with the genes (Fig 2F and S1 Table). The results indicate that single-cell isoform expression data can be used to illustrate cellular heterogeneity and distinguish different types of cells as single-cell gene expression does.

Furthermore, isoforms were found to show different expression patterns compared to the host genes. Isoform switch events largely occurred during the transition from zygote to E2C and from E2C to L2C stages, during the time windows of minor and major ZGA, respectively (Fig 2H and S1 Table). For example, *Cfdp1* increased the gene expression from E2C to L2C, with its isoform *PB.73528.5* showing the same pattern, but the other isoform, *PB.73528.7*, was largely down-regulation at the same stage (Fig 2G). The expression levels of *Cnot7* gene and its isoforms *PB.71257.4* and *PB.71257.165* decreased from E2C to L2C, while *PB.71257.450*, another isoform of the gene was inversely changed (Fig 2G). Gene Ontology (GO) analysis revealed that the genes happened with isoform switch events were enriched in cytoplasmic translation, cell division, mRNA and DNA metabolic processes, etc. (Fig 2I). Functional and structural changes were identified in only 7 genes using Fun-Fam [34,35] (S1 Fig). These results indicate that isoform switch regulation largely exists during embryonic development, especially during the ZGA process, in addition to gene expression level.

**Table 1. Quality evaluation of samples from 3 batches.**

|  | Library | Batch 1 | Batch 2 | Batch 3 |
|---|---|---|---|---|
| Polymerase Reads | Polymerase Reads | 5534582 | 6002833 | 7099439 |
|  | Polymerase Yield (GB) | 427.02 | 540.46 | 410.25 |
|  | Polymerase Max Length | 479619 | 493469 | 483234 |
|  | Polymerase Mean Length | 77155.42 | 90033.59 | 57786.35 |
|  | Polymerase Read N50 | 154554 | 167339 | 130615 |
| Subreads | Subread Yield (GB) | 420.1 | 535.33 | 404.78 |
|  | Subreads Max Length | 479619 | 493469 | 483234 |
|  | Subreads Mean Length | 3001.47 | 4482.57 | 3127.49 |
|  | Subread N50 | 3557 | 5186 | 3508 |
| CCS | CCS Reads | 4412788 | 5079207 | 5914414 |
|  | CCS Yield (GB) | 15.6 | 25.46 | 23.5 |
|  | CCS Max Length | 27871 | 27593 | 27810 |
|  | CCS Mean Length | 3534.33 | 5013.54 | 3973.65 |
|  | CCS N50 Length | 4216 | 5791 | 4760 |
|  | CCS Mean Passes | 27 | 19 | 17 |
|  | CCS MeanQV | 0.96 | 0.97 | 0.94 |
| Full-length (FL) Isoform Detection | All Paired | 11184443 | 20967437 | 11008106 |
|  | FL | 10013148 | 19427495 | 9427600 |
|  | Non-FL | 206358 | 283797 | 283953 |
|  | Unknow | 964937 | 1256144 | 1296552 |
|  | FL (%) | 89.53 | 92.66 | 85.64 |
|  | Non-FL (%) | 1.85 | 1.35 | 2.58 |
|  | Unknow (%) | 8.63 | 5.99 | 11.78 |
|  | FL MeanLen | 934.75 | 888.57 | 1081.15 |
|  | FL N50 | 1128 | 1044 | 1365 |
| Cell Barcode (CB) Identification | FLNC | 10013148 | 19427495 | 9427600 |
|  | CB in Whitelist | 8858087 | 17083701 | 7834317 |
|  | CB in Whitelist (%) | 88.46 | 87.94 | 83.1 |
|  | CB Correction | 254281 | 553029 | 334615 |
|  | CB Correction (%) | 2.54 | 2.85 | 3.55 |
|  | Total Corrected CB | 9112368 | 17636730 | 8168932 |
|  | Total Corrected CB (%) | 91 | 90.78 | 86.65 |

## Isoform diversity decreases along preimplantation embryo development

To explore the connection between gene and isoform expression during mouse preimplantation development, we grouped genes into 6 categories based on the number of isoform types they expressed (S2A Fig). While most genes expressed only 1 type of isoform across different stages, more genes expressed multiple types of isoforms in the earlier stages. In mouse oocytes and zygotes, around 60% of genes expressed more than 1 type of isoform, and nearly 20% of genes were found with over 5 types of isoforms. In contrast, approximately 70% of genes in mESCs expressed only 1 type of isoform, and less than 5% of genes expressed more than 5 types of isoforms (S2A Fig). The same isoform expression characteristics were observed in SCAN-seq data (S2B Fig), indicating a diverse range of isoforms in early mouse embryos [4]. To rule out the possibility that this observation was caused by higher mRNA abundance in early embryos, especially in oocytes and zygotes, we performed oocyte splits. The results showed that the ratios of genes containing different numbers of isoform types were almost consistent among intact oocytes, 1/2 oocytes, and 1/4 oocytes (S2C Fig), suggesting that the

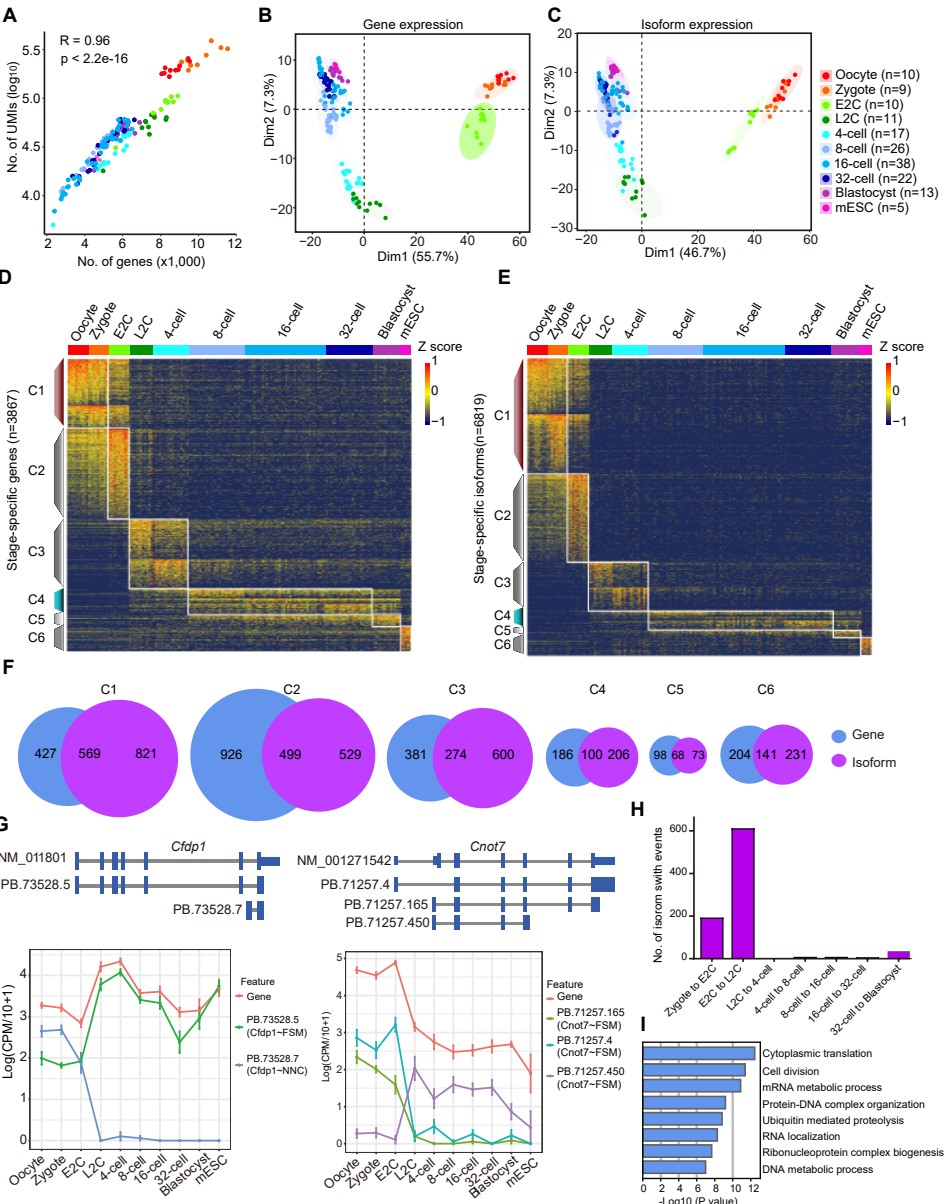

**Fig 2. Gene and isoform expression in the mouse preimplantation embryos.** (A) Number of genes and UMIs detected in each cell at different stages. The raw data for this plot is supplied in S2 Data. (B, C) PCA plot of all the blastomeres and mESCs based on gene expression (B) and isoform expression (C). The PCA loadings are supplied in S1 Table. The raw data for these 2 plots are supplied in S2 Data. (D, E) Heatmap of stage-specific genes (D) and isoforms (E). The raw data for these 2 plots are supplied in S2 Data. (F) Venn plot of pairwise groups of stage-specific genes and isoform-corresponding genes. (G) Isoform switch during embryo development, the upper picture is the reference transcript and our sequence isoform and the lower part is the gene and main isoform expression of *Cfdp1* and *Cnot7*. The raw data for this plot is supplied in S2 Data. (H) Number of isoform switch events between each adjacent embryonic stages. The raw data for this plot is supplied in S2 Data. (I) GO results of genes showed isoform switch during preimplantation embryo development. The raw data for this plot is supplied in S2 Data. GO, Gene Ontology; mESC, mouse embryonic stem cell; PCA, principal component analysis.

detected isoform diversity was hardly affected by the amount of mRNAs. Additionally, the genes expressing more types of isoforms were detected with higher expression levels in both our data and SCAN-seq data (S2D and S2E Fig). To validate this hypothesis, we randomly selected 3 highly expressed genes (CPM > 100) and 3 lowly expressed genes (CPM < 10) in mESC to confirm their isoform diversity by reverse transcription and PCR (RT-PCR). Although there were more types of isoforms revealed by RT-PCR than the sequencing results, the highly expressed genes still showed higher isoform diversity (S2F Fig). We then assessed the isoform dominant level in each gene expressing multiple types of isoforms by calculating the ratio of the UMI number of the major isoform to the total UMI number of the corresponding gene. The major isoform ratios increased from early to late embryonic stages, especially after the ZGA process (S2G Fig). In comparison, the major isoforms accounted for 90% of most genes in mESCs, indicating a dominant isoform expression pattern and less isoform diversity in these cells.

## Large abundance of 3-prime partial transcripts are observed in mouse oocytes and zygotes

Based on the putative integrity of corresponding open reading frames (ORFs), the transcripts were categorized into 5 types: complete isoforms encoding the full ORFs, 3-prime partial transcripts and 5-prime partial transcripts lacking the stop codon and start codon sections respectively, internal transcripts predicted with proteins lacking both ends, and others where the detected ORF lengths in the isoforms were below the software-set threshold [36] (Fig 3A). When compared to the annotated transcription start site (TSS), the 5-prime partial transcripts exhibited the lowest overlap ratio with the CAGE peaks (S3A Fig), indicating that some of these transcripts might be generated by incomplete reverse transcription.

As anticipated, the complete transcripts displayed the longest lengths, while the internal transcripts were the shortest (S3B Fig). However, the predicted protein length was similar for the 3 incomplete transcript types (S3C Fig). Intriguingly, we observed that the 3-prime partial transcripts were highly expressed in oocyte and zygote, but their expression dramatically decreased from the E2C stage (Fig 3B and 3C). This expression pattern was also observed in the SCAN-seq data (S3D and S3E Fig). Subsequently, we conducted GO analysis on genes detected with 3-prime partial transcripts (S2 Table). These genes were enriched in pathways related to RNA processing, cell cycle checkpoint, ribonucleoprotein complex biogenesis, DNA metabolic process, chromatin organization, etc. (Fig 3D), all of which are known to play essential roles in mouse and human preimplantation embryo development [13,16,37–40].

We then selected some candidates to validate the enrichment of 3-prime partial transcripts. The host genes related to RNA processing (*Sf3b2*, *Srpk1*) and protein translation and transporting (*Dnajc3*, *Hsp90aa1*) were revealed by gel analysis of mouse oocyte RT-PCR products (Fig 3E). Furthermore, no stop codons were identified in these transcripts according to the Sanger sequencing results (Fig 3F).

## Expression of *Ncl* showed significantly isoform switch during embryonic development

The Nucleolin gene encodes NCL, which is involved in various cellular processes such as ribosome biogenesis, chromatin organization and stability, and DNA and RNA metabolism [41]. It also regulates totipotent genes expression with KAP1 and LINE1 [27,28]. Isoform sequencing revealed that the *Ncl* gene encodes 6 isoform types, categorized based on the number of RNA recognition motifs (RRMs) they contain (Fig 4A). RT-PCR showed that the short isoform (*Ncl-S-350*) was more abundant than the complete isoform (*Ncl-FL-71*) in mouse oocyte

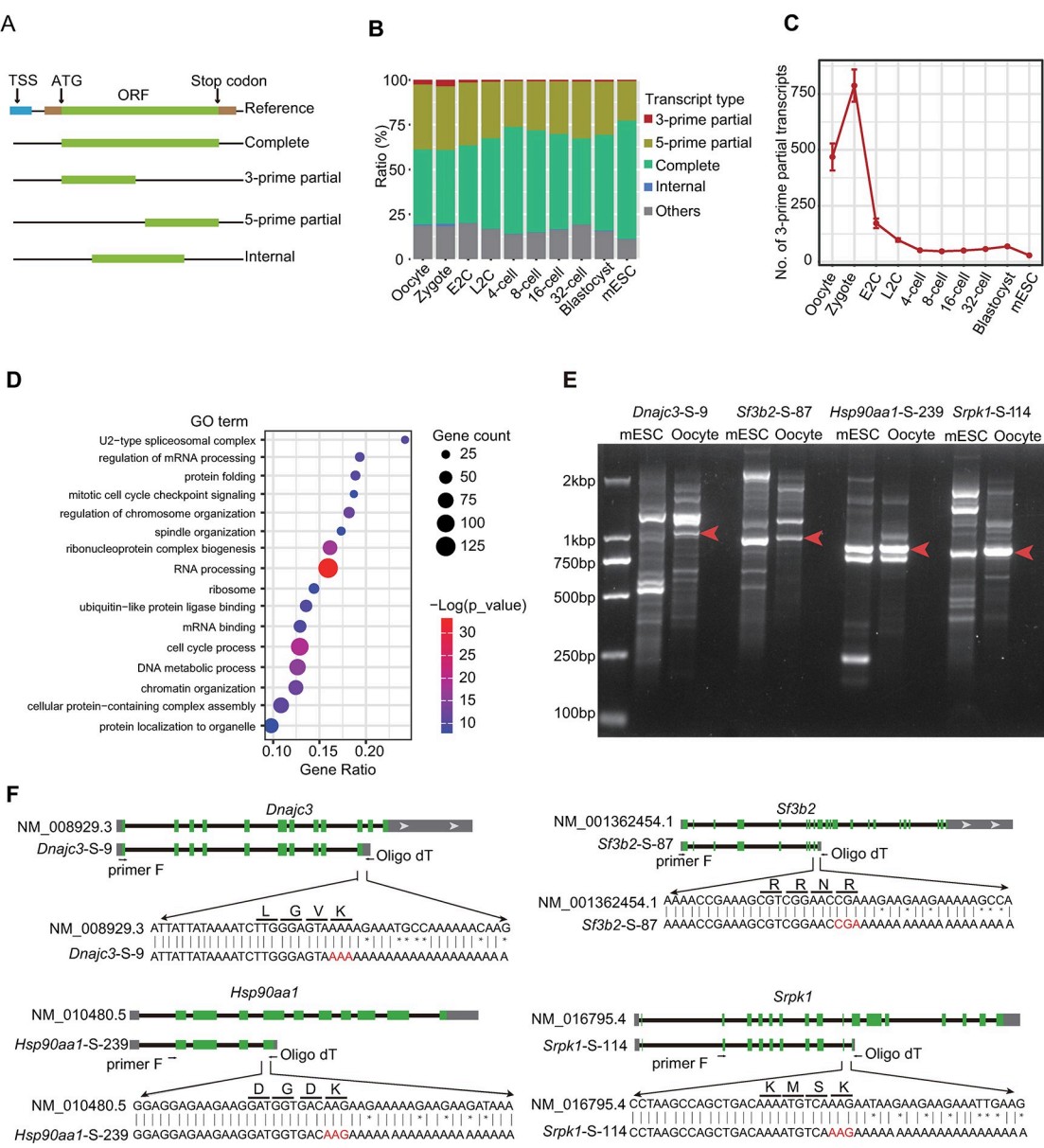

**Fig 3. Expression patterns of different types of transcripts during mouse preimplantation embryo development.** (A) Schematic diagram of different transcript types. (B) Ratios of each type of transcripts at different stages. The raw data for this plot is supplied in S3 Data. (C) Number of 3-prime partial transcripts detected at each stage. The raw data for this plot is supplied in S3 Data. (D) The GO analysis of 3-prime partial transcripts. The raw data for this plot is supplied in S3 Data. (E) Gel picture showing the isoforms by RT-PCR of *Dnajc3*, *Sf3b2*, *Hsp90aa1*, and *Sprk1* in mouse oocytes and mESCs. The raw image for this plot is supplied in S1 Raw Images. (F) Sanger sequencing of the candidate 3-prime partial transcripts in Fig 3E. GO, Gene Ontology; mESC, mouse embryonic stem cell.

(Fig 4B). The 2 of most enriched short isoforms were confirmed as 3-prime partial isoform by Sanger sequencing (Fig 4C). *Ncl* abundance was first down-regulated in the E2C stage and then increased from the L2C stage at the gene level (Fig 4D). The 2 categories of short *Ncl* isoforms were highly expressed in oocytes and zygotes and then almost disappeared. Conversely, the complete *Ncl* isoform showed lower expression in maternal RNA and was largely up-regulated after ZGA (Fig 4D). Our findings highlight the isoform switch of the *Ncl* gene during the ZGA process, which is masked in gene-level analysis.

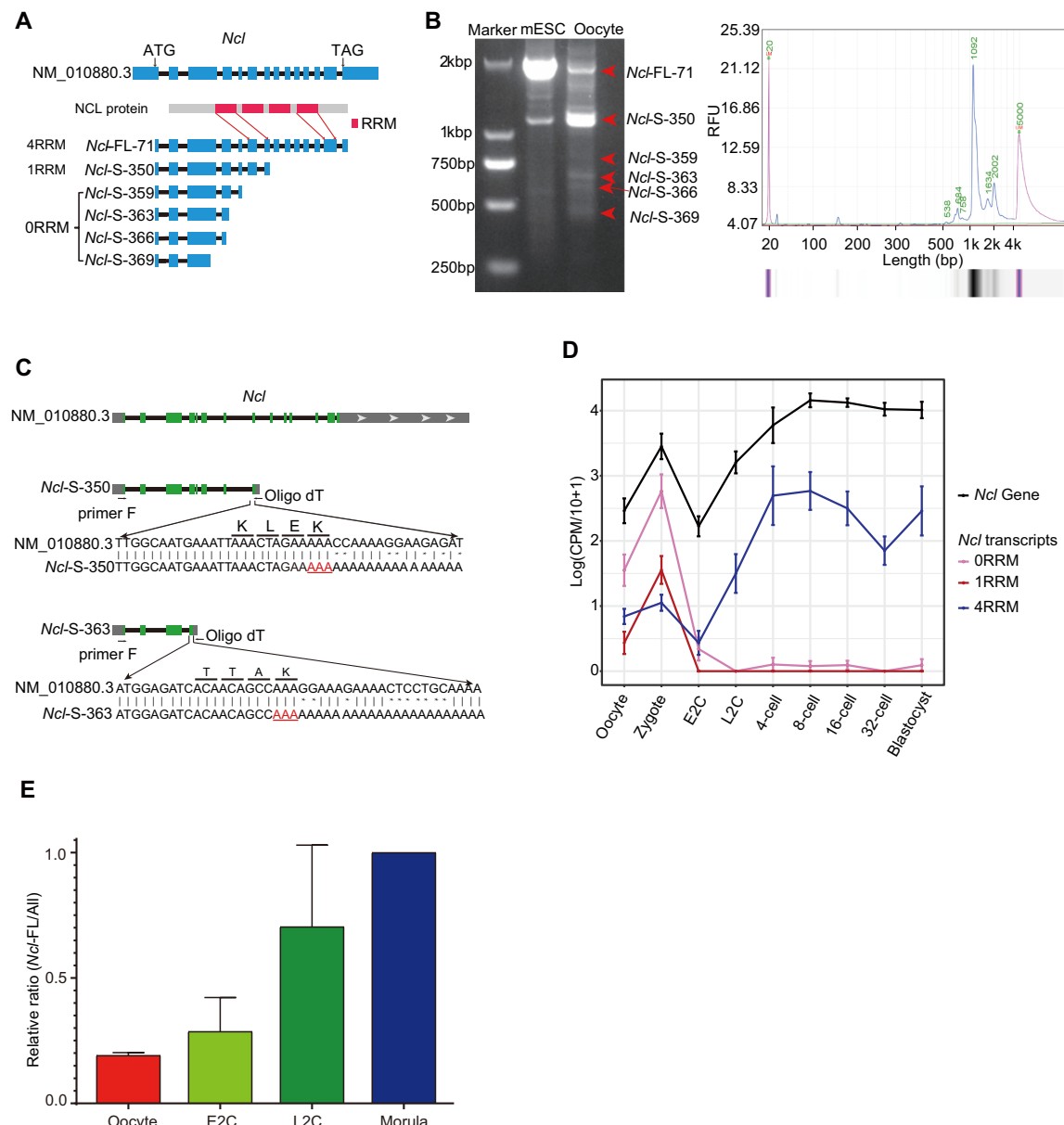

**Fig 4. Isoform expression pattern of *Ncl* during mouse preimplantation embryonic development.** (A) Schematic diagram of *Ncl* isoforms. (B) Gel picture (left) and Q-sep result (right) of *Ncl* RT-PCR products from mouse oocyte. The raw image for this plot is supplied in S1 Raw Images. (C) The Sanger sequencing results of the top 2 enriched short *Ncl* isoforms. (D) Expression levels of each category of *Ncl* isoforms and the *Ncl* gene at different stages. The raw data for this plot is supplied in S4 Data. (E) The relative ratios of *Ncl* full-length isoforms at different stages detected by RT-qPCR. The relative ratio of *Ncl*-FL/All in morula was set as 1.0. The morula had 1 sample, while other stages all had 3 replicates. The raw data for this plot is supplied in S4 Data.

To demonstrate this result, we performed reverse transcription and real-time quantitative PCR (RT–qPCR) using primers targeting all the *Ncl* isoform types or only the complete type, respectively, and calculated the relative percentages of the full-length isoform in different stages of mouse preimplantation embryos. As expected, less than 20% of the *Ncl* transcripts are complete in oocytes when we set the full-length relative ratio as 100% at the morula stage (Fig 4E). This result confirmed the dynamic isoform switch during embryo development in vivo.

## TEs are dynamically activated during embryonic development

Due to the repetitive and interspersed features of TE sequences and their transcripts, TGS-based sequencing is more suitable for TE research [42,43]. Our long-read and highly accurate results can help us investigate TE expression at specific loci. TE expression was quantified in each single cell. Generally, the amount of TE RNAs belonging to different super-families decreased along embryonic stages (Fig 5A). Although maternal RNA contains the largest pool of TE elements, zygotes were detected with more TE RNA copies, suggesting TE as an important regulator to promote minor ZGA (Fig 5A). Additionally, TE expression elevated from 32-cell to blastocyst stage, indicating that TEs play a role in embryonic pluripotent stem cells. Our single-cell direct isoform sequencing data enabled mapping the TE reads to specific loci confidently, and we also calculated the number of expressed TE loci at each stage. Hundreds of TE loci were transiently transcribed at the zygote stage, further supporting the deduction that TEs regulate minor ZGA (Fig 5B). More active TE loci were detected in blastocysts than morulae, also indicating the important role of TEs in embryonic pluripotent stem cells (Fig 5B).

We further calculated the TE expression level according to total reads or single locus mapped reads belonging to each TE superfamily in each single cell (Fig 5C). In both calculation ways, the oocyte and zygote were detected with higher expression levels. However, more different expression patterns were observed between total TE superfamily expression and individual TE locus expression. For example, LINE was slightly down-regulated from E2C to 32-cell stage when looking at the superfamily, but the transcription level at each locus was up-regulated. Differently, the total LTR expression transiently increased at L2C stage, but each locus was detected with lower expression level (Fig 5C). These differences still exist when calculating in each TE family (S4A and S4B Fig). The ERVL, which has been proven to be associated with totipotent genes' activation [18,25,26], is indeed the highest expressed in L2C samples. However, each active ERVL site in the L2C genome did not express the most copies of corresponding RNAs. This indicates that more ERVL sites are transiently active to regulate a large scale of major ZGA genes at L2C stage (S4A and S4B Fig). Gaining information on locus-specific TE expression may help us to gain a deeper understanding of how TEs regulate different developmental processes.

To study the role of specific TEs at distinct locations, not categorized in a family or subfamily, in regulating preimplantation embryo development, we sought out TE loci with stage-specific expression patterns. We identified a total of 3,894 TE loci, which could be classified into 5 clusters based on their expression patterns across all embryonic stages (Fig 5D and S3 Table). Specifically, Cluster 1 (C1) TEs exhibited higher expression in oocytes and zygotes. Cluster 2 (C2) TEs were predominantly expressed in the E2C stage. Cluster 3 (C3) and Cluster 4 (C4) TEs showed high expression in the L2C to 4-cell stages and 8-cell to blastocyst stages, respectively. The mESC-specific TEs were grouped in Cluster 5 (C5).

Subsequently, we explored the involvement of ERVL and LINE1 subfamilies in the MZT process and later stage development [18,25–27] (Figs 5E, 5F, S4C and S4D). As anticipated, MERVL-int and MT2_mm, which only contain the LTR promoter of MERVL element, were the primary active MERVL subtypes in C3, a stage during which major totipotent genes are expressed [19,44] (Fig 5E). Conversely, MT2B1 was the most active subtype in the maternal genome. The LINE1 superfamily has been reported to silence totipotent genes such as *Dux* [27]. In our data, Lx7 emerged as the most active LINE1 subfamily since the L2C stage (Figs 5F and S4D).

A TE subfamily consists of hundreds to thousands of TE copies from different loci, and these copies are transcribed independently. We observed diverse activation of different TE copies even within the same subfamily (S5A and S5B Fig). For example, although MERVL-int and MT2_mm participate in ZGA, some copies from chromosome 1 and chromosome 5 were

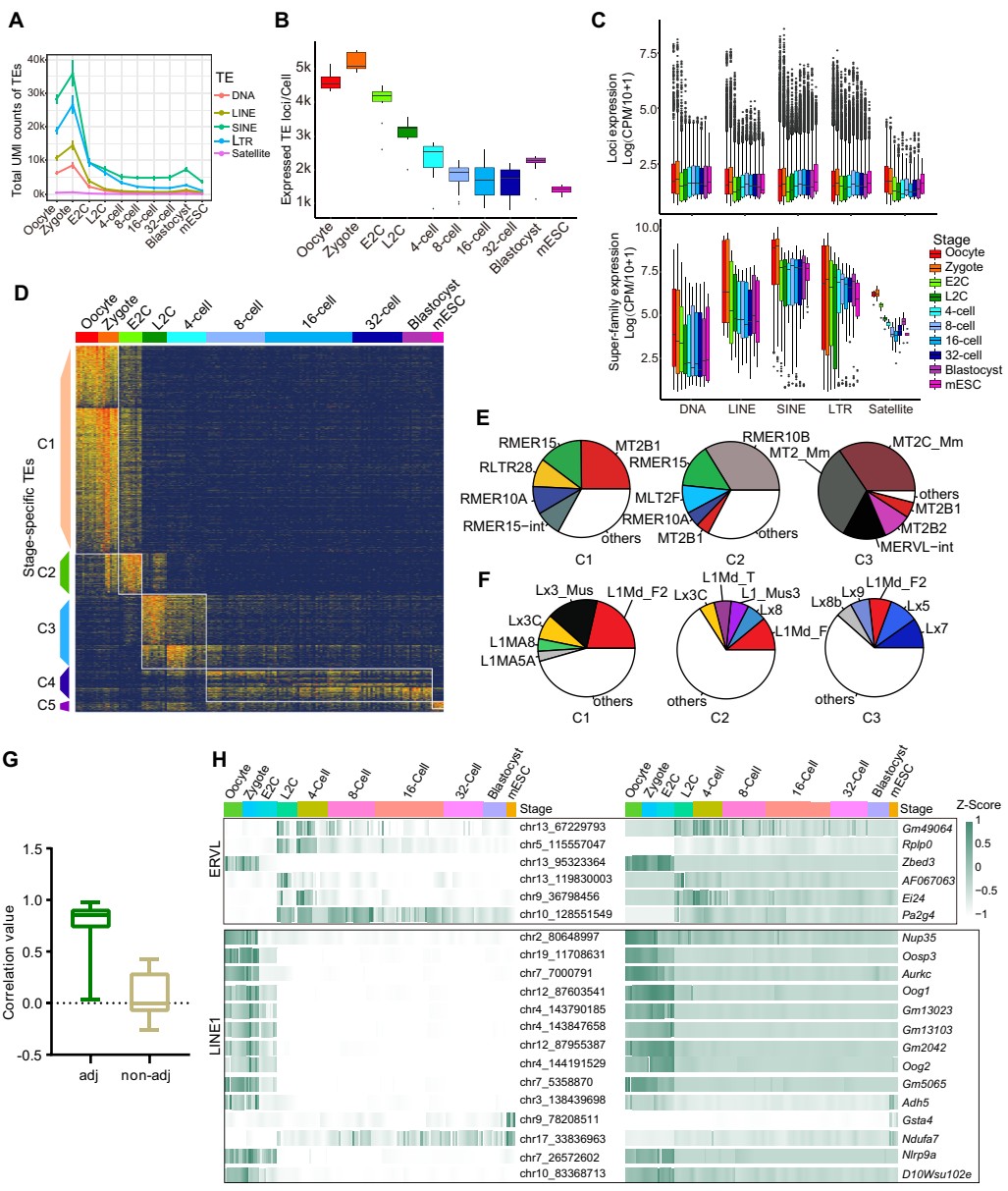

**Fig 5. Dynamic expression patterns of TEs during mouse preimplantation embryo development.** (A) UMI counts of each TE superfamily detected in single cells at each stage. The raw data for this plot is supplied in S5 Data. (B) Number of expressed TE loci per cell at each stage. The raw data for this plot is supplied in S5 Data. (C) Expression level of each TE locus (upper panel) and all TE counts (lower panel) belonging to different classes in each stage, satellite was also included. Each dot represents a cell. Only active loci in each cell were calculated. The raw data for this plot is supplied in S5 Data. (D) Expression heatmap of stage specific TEs (*n* = 3,894 unique loci). The TEs were ordered by their averaged expression levels within each stage (also see S3 Table). The raw data for this plot is supplied in S5 Data. (E) The top5 subfamilies of ERVL expressed in clusters C1–C3 in Fig 5D. (F) The top5 subfamilies of LINE1 expressed in clusters C1–C3 in Fig 5D. (G) Correlation values between expression levels of representative stage-specific LINE1 and ERVL loci (averaged UMI counts >5) and their adjacent genes (<10 kb distance). Non-adjacent genes are defined as genes over 1 Mb away from the TE loci. The raw data for this plot is supplied in S5 Data. (H) Heatmap showing examples of consistent expressions between specific TE loci and their adjacent genes along developmental stages. The raw data for this plot is supplied in S5 Data. LINE, long interspersed element; TE, transposable element.

exclusively active in the maternal genome (S5A Fig). Most of the LINE1 copies belonging to different subfamilies were actively transcribed in oocytes, but more Lx7 copies became active only after the E2C stage (S5B Fig). The expression level of TE loci showed a notably higher correlation with that of adjacent genes (Fig 5G). We observed consistent expression of specific LINE1 and ERVL loci with adjacent genes, indicating common active regulations for different TE families (Fig 5H). As TEs can function as transcriptional regulatory elements such as enhancers, detecting locus-specific TE expression provides more insights into understanding the regulatory mechanism of preimplantation embryo development [45].

## Discussion

Single-cell sequencing has significantly advanced our research in preimplantation embryo development. We have identified genes, isoforms, and TE elements that are specific to different developmental stages, revealing a rich diversity of isoforms in the maternal contents. By modifying the HIT-scISO-seq to a low-throughput method, we were able to track cells from the same embryo during the experiment. This single-cell approach not only illustrated differences across developmental stages but also highlighted cell heterogeneity within a specific stage, allowing for the evaluation of fate differentiation of blastomeres within an embryo.

The 2 blastomeres in the 2-cell embryo showed high consistency in both gene and isoform expression levels, but became increasingly different from each other from the 4-cell stage (S5C and S5D Fig), that fate divergence could be revealed at the 4-cell stage on the transcriptional level. When comparing the heterogeneity between cells from different embryos at the same stage, the correlation values also decreased along with developmental stages. The decreased cell–cell correlation at the zygote stage likely resulted from batch effects of different embryo replicates. However, the low-throughput approach, affected by the low-throughput and high cost of TGS, also makes it difficult to obtain a large sample size, especially since the cells collected in the blastocyst stage are insufficient to illustrate intra-embryo heterogeneity. We believe that future studies using high-throughput approaches are more powerful in fully specifying the transcript regulations in cell fate specification at the blastocyst stage.

TGS-based isoform sequencing, at the bulk or single-cell level, has annotated an abundance of novel transcripts and splicing events in preimplantation embryos [4,17,22]. However, it remains unclear how different types of isoforms regulate the developmental process. In the present study, by dividing the transcripts into subtypes according to their coding characteristics, we found a large number of 3-prime partial transcripts, which lack stop codons, in mouse MII oocyte and zygote (Fig 3B, 3C and 3E). This type of transcript has been extensively studied in cancer and is considered an oncogenic factor [46]. In early embryos, these transcripts might be important for the MZT process, as the host genes are highly enriched in biological processes responsible for mouse and human preimplantation embryo development (Fig 3D). Further studies are needed to resolve the generation and function of these transcripts.

TEs are the main components of the mammalian genome. However, their biological function is still largely unclear, and most of them were previously regarded as parasites or "junk DNA" [27]. Although some TE classes have been investigated in preimplantation embryos, previous studies almost exclusively used NGS-based analysis, only revealing the TEs at the subfamily level [22,27,47,48]. Our long-read direct isoform sequencing directly quantifies the TE transcription from different loci (Figs 5 and S3). Our data displays more detailed TE expression dynamics, which helps us to investigate these genome "dark matters" in more detail.

TEs had been reported to regulate gene expression by different ways. For example, LINE1 has the ability to increase the chromatin accessibility [21,27,49], while MERVL and MT2_mm can derive totipotent gene expression in 2-cell and 4-cell embryo [18,25,26]. Therefore,

exploring how dynamic expression of locus-specific TEs regulates gene expression requires further investigation. On the other hand, abnormal expression of TEs in most differentiated tissues is harmful to humans. For instance, LINE1 overexpression is highly related to cancers such as gastric cancer and lung squamous cell carcinoma [50–52]. HERVK overexpression is related to aging in mouse, monkey, and human [53]. Therefore, measuring TE expression at the locus-specific level offers a new way to decode the mechanisms in various human diseases.

Beyond just TE expression, several studies have identified transcript isoforms where TEs are used as alternative promoters for gene expression [18,54–57]. We also attempted to find TE chimeric transcripts in the mouse preimplantation embryos. A total of 6,143 TE chimeric transcripts were identified, with over 80% only detected in 1 cell with 1 copy (S4 Table). One reason could be the low expression levels of these transcripts. Considering the limited sequencing depth from the TGS in this study, it is difficult to fully detect these transcripts. Another reason might be the current bioinformatic methods, which are not suitable for analyzing the TE chimeric transcripts. As the recent study by Berrens and colleagues [22] also captured quite a limited number of TE-derived isoforms in each cell (approximately 200 for the mouse 2-cell sample and approximately 20 for human iPSCs). The increase of TGS throughput and the development of bioinformatics would help us to further explore on such interesting regulations.

## Materials and methods

### Ethics statement

All animal experiments were performed according to the guidelines of the Institutional Animal Care and the Ethics Committee of the Guangzhou Institutes of Biomedicine and Health (Guangzhou, China). The research license number is IACUC2020113.

### Animals and single blastomere collection

We used 6- to 8-week-old C57BL/6J female mice and DBA/2NCrl male mice in the experiment. The female mice were first injected with 7.5 IU of pregnant mare's serum gonadotropin (PMSG) (Ningbo SanSheng Biological Technology, Cat. 110044564) and with 7.5 IU of human chorionic gonadotropin (hCG) (Ningbo SanSheng Biological Technology, Cat. 50030248) after 46 to 48 h injected. After mating, the embryos of each stage were collected at defined time periods after hCG administration [58]: 20 h (MII oocyte, no mating), 22 to 24 h (zygote), 30 to 32 h (early 2cell), 46 to 48 h (late 2cell), 54 to 56 h (4cell), 68 to 70 h (8-cell), 78 to 80 h (16cell to 32cell), and 88 to 90 h (early blastocyst). All animal experiments were performed according to the guidelines of the Guangzhou Institutes of Biomedicine and Health (Guangzhou, China). Collection of single blastomeres at each stage was carried out as previously described [4].

### Single-cell cDNA amplification and TGS library construction for PacBio sequencing

We used the same amplification procedure as SCAN-seq [4], except for changing the reverse transcription primer with a 10× gel bead for each reaction for the embryonic samples. Then, each pre-amplification product was purified by 0.6× Ampure XP beads (Beckman, Cat. A63882). The concentration was measured using Qubit dsDNA HS and BR Assay Kits (Invitrogen, Cat. Q32854). The PCR product from about 60 blastomeres which were confirmed of effective amplification were pooled together in proportion to the number of amplified cycles. We took 100 ng of the pooled cDNA to build PacBio sequencing library following the protocol of HIT-scISOseq [6] and sequenced for 1 cell with HiFi mode.

## Cell barcode sequence identification by Sanger sequencing

About 2 ng of the pre-amplified cDNA of each cell was further amplified using 2 × Taq Plus Master Mix (Vazyme, Cat. P212), and then cloned into T vector (Transgen, Cat. CT111-01). Next, the ligated plasmid transferred into *Trans5α* chemically competent cell (Transgen, Cat. CD201-01) by heat shock. The M13 primer were used to identify positive clones inserted with cDNA fragments. Single clones of bacteria were collected for Sanger sequencing to identify the barcode sequence of each cell.

## Mouse ES cell culture

Mouse E14 Tg2A (E14) ES cells (male) were used for all experiments. The mESCs were cultured on 0.1% gelatin-coated plates in ES-FBS culture medium as previously described [27].

## Validation of the 3-prime partial transcripts

The oocytes or the mESCs RNA were reverse transcription using oligo-dT primer (AAGCAGTGGTATCAACGCAGAGTACTTTTTTTTTTTTTTTTTTTTTTTTTT). Then, the anchor sequence of oligo-dT primer and a primer located in the start codon of the interested gene (*Ncl*: ATGGTGAAGCTCGCAAAGGC; *Dnajc3*: ATGGTGGCCCCCGGCTCGGTG; *Sf3b2*: ATGGCGGCGGAGCATCCCGAACCT; *Hsp90aa1*: ATGCCTGAGGAAACCCAGACCCA; *Srpk1*: ATGGAGCGGAAAGTGCTCGCGCT) were used to amplify all isoforms containing the 5′ sites. The PCR products were first checked on 1.5% agarose gel. The candidate gel bands were recovered and the sequences were confirmed by Sanger sequencing.

## Validation isoform diversity of highly and lower expressed genes

The mESCs RNA were reverse transcription using oligo-dT primer as previous mentioned. Then, the forward primer that could distinguish most diverse isoforms was used to amplification target genes (*Srsf7*: ATGTCACGCTACGGGCGGTA; *Rps5*: CTGTCTGTATCAGGGCGGCG; *Rps19*: TTTCCCCTGGCTGGCAGCGC; *Plp2*: ATGGCGGATTCTGAGCGTCT; *Mrps6*: ATGCCCCGCTACGAGTTGGC; *Ss18l2*: ATGTCTGTCATCTTCGCTCCTG). All the reverse primer used was oligo-dT. The PCR products were checked on 1.5% agarose gel.

## Single-cell isoform sequencing data processing

We used stand-alone versions of SMRT-Link (version 8.0.0.80529) software package to transform raw Subreads to Calling Circular Consensus Sequencing (CCS) reads with the following parameters: "—min-passes 0—min-length 50—max-length 21000—min-rq 0.75." After CCS calling, we used HIT-scISOseq analysis software kit scISA-Tools (https://github.com/shizhuoxing/scISA-Tools) for Full-Length Non-Concatemer (FLNC) reads identification, cell barcode and UMI extraction and correction.

## Alignment and generation of single-cell gene expression matrix

After trimming the primers, cell barcodes, UMIs, and polyA tails, the remaining FLNC sequences were aligned to mouse genome (10x Genomics pre-build mouse mm10 reference dataset: refdata-gex-mm10-2020-A) using minimap2 (version 2.17-r974-dirty) in spliced alignment mode with the following parameters: "-ax splice -uf—secondary = no -C5." Then, we used gffcompare (version 0.11.6) to assign the mapped FLNCs to mm10 annotation gene models (10x Genomics pre-build mouse mm10 reference dataset: refdata-gex-mm10-2020-A) base on FLNCs genome alignment SAM file. Next, based on the identified cell barcodes, we

used scGene_matrix utility of scISA-Tools to generate the single-cell gene expression matrix. The expression values were normalized as copy number per 100,000 mapped reads (CPM/10).

### SIRV data evaluation

The FLNC reads were aligned to the SIRVome using minimap2 (version 2.17-r974-dirty) with the following parameters: "-ax splice -uf—MD—sam-hit-only." We only annotated the reads with assigned barcodes and valid UMIs. Then, we used gffcompare (version 0.11.6) to assign the mapped FLNCs to SIRVome annotation GTF (SIRV-Set4) base on FLNCs SIRVome alignment SAM file. A confusion matrix was generated with the counts of FLNCs assigned to the primary SIRV isoforms or not using an in-house script.

### Nonredundant isoforms classification and quality assessment

First, the "collapse_isoforms_by_sam.py" python script in cDNA_Cupcake software package (https://github.com/Magdoll/cDNA_Cupcake) was used to collapse mapped FLNCs to nonredundant isoforms with parameters: "—dun-merge-5-shorter." After that, we used SQANTI3 (https://github.com/ConesaLab/SQANTI3). To assess whether transcripts are within known TSSs, we aligned them using the CAGE peak data (mouse.refTSS_v3.1.mm10.bed) provided with the "—CAGE_peak" parameter in SQANTI3. We further used SQANTI3 "RulesFilter" script to filtered artifact isoforms. Isoforms which classified as FSM, ISM, NIC, and NNC were kept for downstream analysis.

### Isoform type classification by ORF prediction

After being processed with SQANTI3, the FASTA file of mapped genome sequences were extracted according to the SQANTI3 output GTF file. Base on the FASTA file, we used TransDecoder (v5.5.0) for ORF extraction and prediction. For those predicted with multiple ORFs, the longest ones were selected as the representative ORF. TransDecoder assigns each detected isoform as one of 4 types based on whether then contains the start and stop codon of the reference ORF: complete, 5-prime partial, 3-prime partial, and internal. Additionally, we assigned the isoforms that did not mapped to an ORF region by TransDecoder as others.

### Generation of single-cell isoform expression matrix

After the SQANTI3 procedure, the scIsoform_matrix utility of scISA-Tools was used to generate single-cell isoform expression matrix based on the identified cell barcodes. We further filtered isoforms detected in less than 5 cells and finally 68,012 isoforms in the mouse embryonic samples were preserved.

### PCA analysis based on gene and isoform expression

Before PCA dimensionality reduction, we used "FindVariableGenes()" function in Seurat R package to select the top 1,000 highly variable genes and isoforms, respectively. Then, the "PCA()" function in FactoMineR was used for dimension reduction process and we used the function "fviz_pca_ind()" in factoextra R package to plot the PCA map.

### Stage-specific genes and isoforms

Based on the gene expression matrix and isoform expression matrix, respectively, we used "edgeR" to find differentially expressed genes/transcripts between each pair of adjacent embryonic stages under the criterion of logFC>1 and $p$-value <0.01. A total of 3,867 and 6,819 stage-specific genes and transcripts were identified, respectively. These genes and transcripts

were clustered into 6 groups according to their expression patterns across all stages. Visualization of these genes' and transcripts' expression was done using R package "pheatmap."

### SCAN-seq data processing

We downloaded the SCAN-seq data available from the Sequence Read Archive (SRA) database (accession number: PRJNA616184). Following the described data processing steps of SCAN-seq. Briefly, nanoplexer (https://github.com/hanyue36/nanoplexer/) was used to demultiplex barcode for each cell in the library, and nanofilt (v2.5.0) was used for filtering low-quality reads (qscore <7) and short reads (length <100 bp), then Pychopper (v2.3) (https://github.com/nanoporetech/pychopper) was used to extract full-length reads.

After obtaining the full-length reads, we generated the gene expression matrix and isoform expression matrix using the same procedure as we did for HIT-scISOseq data.

### Isoform switch analysis

Based on the isoform expression matrix, we used "IsoformSwitchAnalyzeR" to identify switch isoforms between each pair of adjacent embryonic stages under the criterion of isoform_switch_q_value<0.05 and gene_switch_q_value<0.01.

### TE expression analysis

To quantify TE at the locus level, we first aligned all FLNC (full-length non-chimeric) reads to the mm10 genome. Based on the TE annotation file obtained from UCSC (http://hgdownload.soe.ucsc.edu/goldenPath/mm10/database/rmsk.txt.gz), we calculated the overlap between each uniquely aligned FLNC read and TE loci. Subsequently, we applied filtering criteria: the starting or ending position of FLNC alignment to the genome must fall within an overlapping TE locus, and only the TE locus with the longest overlap length was considered for quantitative counting of the same FLNC.

Following the above steps, we performed aggregate counting based on the cell barcode sequences corresponding to each FLNC, enabling us to obtain quantitative expression measurements of TE loci at the single-cell level.

To quantify the expression of TE-associated chimeric transcripts, we developed an in-house script. Firstly, this script extracts the chimeric alignments from FLNC. Secondly, it employs a method to identify TE sites that overlap with FLNC, as described in the unique TE mapping method above. Thirdly, we align the chimeric-mapped FLNC with the protein-coding gene positions from the reference annotation, aggregate these alignment results, and thus determine the expression quantification (UMI counts) of protein-coding genes linked to each TE locus.

## Supporting information

**S1 Fig. Isoform switch caused functional/structural changes of genes.** Expression pattern of gene isoforms which showed switch during preimplantation embryo development. The exact functional/structural domains predicted in each isoform are annotated based on the CATH database (http://cathdb.info/search/by_sequence).
(EPS)

**S2 Fig. Relationship between gene and isoform expression.** (A, B) The ratios of genes detected with different numbers of isoform types for each stage of mouse embryos and mESCs in this study (A) and SCAN-seq data (B). The raw data for these 2 plots are supplied in S6 Data. (C) The ratios of genes detected with different numbers of isoform types for full oocyte, 1/2 oocyte, and 1/4 oocyte. The raw data for this plot is supplied in S2 Data. (D, E) Expression levels of genes

detected with different numbers of isoform types for each stage of mouse embryos and mESCs in this study (D) and SCAN-seq data (E). The raw data for these 2 plots are supplied in S6 Data. (F) Gel view of cDNA amplification products of each gene. *Srsf7*, *Rps5*, and *Rps19* are examples of highly expressed genes (CPM >100) and *Plp2*, *Mrps6*, and *Ssl8l2* are lowly expressed genes (CPM <10). The raw image for this plot is supplied in S1 Raw Images. (G) Density plot showing the proportion of the major isoforms in genes expressing multiple isoform types. Only genes detected with UMI counts over 5 were included. The raw data for this plot is supplied in S6 Data.
(TIF)

**S3 Fig. The characteristics of different types of transcripts.** (A) Ratios of the transcripts overlapped with annotated TSS. Transcripts with the 5-terminal locating within 200 bp of the CAGE peaks are regarded as overlapped transcripts. The raw data for this plot is supplied in S7 Data. (B) Length distribution of different types of transcripts. The raw data for this plot is supplied in S7 Data. (C) Relative length of the predicted protein to the complete reference ORF of each type of transcript. The raw data for this plot is supplied in S7 Data. (D) Ratios of each type of transcript at different stages calculated using SCAN-seq data. The raw data for this plot is supplied in S7 Data. (E) Expression level of the 3-prime partial transcripts detected at each stage in SCAN-seq data. The raw data for this plot is supplied in S7 Data.
(EPS)

**S4 Fig. Characteristics of TE expression during preimplantation embryo development.** (A) Expression level of all TE counts belonging to different families in each stage. Each dot represents a cell. The raw data for this plot is supplied in S5 Data. Expression level of each TE locus (lower panel) belonging to different families in each stage. Only active loci in each cell were calculated. Each dot represents a cell. The raw data for this plot is supplied in S5 Data. (B) The top 5 subfamilies of ERVL expressed in clusters C4–C5 in Fig 5E. (C) The top 5 subfamilies of LINE1 expressed in clusters C4–C5 in Fig 5F.
(EPS)

**S5 Fig. The loci expression of TEs and cell heterogeneity along developmental stages.** (A) The loci expression of stage specific TEs (mean expression >1 log count across all cells) belonging to ERVL family. The raw data for this plot is supplied in S8 Data. (B) The loci expression of stage specific TEs (mean expression >1 log count across all cells) belonging to LINE1 family. The raw data for this plot is supplied in S8 Data. (C) Correlation coefficients of blastomeres within the same embryos or different embryos at the same stage base on gene expression data. The raw data for this plot is supplied in S8 Data. (D) Correlation coefficients of blastomeres within the same embryos or different embryos at the same stage base on isoform expression data. The raw data for this plot is supplied in S8 Data.
(EPS)

**S1 Table. The list of stage-specific genes and stage-specific isoforms.**
(XLSX)

**S2 Table. The expression matrix of 3-prime partial transcripts.**
(XLSX)

**S3 Table. The expression matrix of TE loci.**
(XLSX)

**S4 Table. The list of TE chimeric genes.**
(XLSX)

**S1 Raw Images. Raw images.**
(PDF)

**S1 Data. Raw data for Fig 1.**
(XLSX)

**S2 Data. Raw data for Fig 2.**
(XLSX)

**S3 Data. Raw data for Fig 3.**
(XLSX)

**S4 Data. Raw data for Fig 4.**
(XLSX)

**S5 Data. Raw data for Figs 5 and S4.**
(XLSX)

**S6 Data. Raw data for S2 Fig.**
(XLSX)

**S7 Data. Raw data for S3 Fig.**
(XLSX)

**S8 Data. Raw data for S5 Fig.**
(XLSX)

## Acknowledgments

We thank Man Zhang (Guangzhou Laboratory) for teaching the methods to collect the preimplant embryos from mice. We thank Boyan Huang (Guangzhou Laboratory) for providing with the mESC cells. We thank Xining Li and Enze Deng (Guangzhou Laboratory) for helping analysis of some preliminary RNA-seq data.

## Author Contributions

**Conceptualization:** Chuanle Xiao, Xiaoying Fan.

**Data curation:** Zhuoxing Shi, Xiaoying Fan.

**Formal analysis:** Zhuoxing Shi.

**Funding acquisition:** Xiaoying Fan.

**Investigation:** Chaoyang Wang, Qingpei Huang, Lei Chang, Xiaoying Fan.

**Methodology:** Zhuoxing Shi, Qingpei Huang, Rong Liu, Dan Su, Lei Chang.

**Project administration:** Chuanle Xiao, Xiaoying Fan.

**Resources:** Chaoyang Wang, Zhuoxing Shi, Rong Liu.

**Software:** Zhuoxing Shi.

**Supervision:** Chuanle Xiao, Xiaoying Fan.

**Validation:** Chaoyang Wang, Qingpei Huang, Dan Su.

**Visualization:** Zhuoxing Shi.

**Writing – original draft:** Chaoyang Wang, Zhuoxing Shi, Lei Chang, Xiaoying Fan.

**Writing – review & editing:** Chaoyang Wang, Zhuoxing Shi, Xiaoying Fan.

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
