## [Editor Report · Decision Letter 0]

25 Jun 2023

Dear Dr Fan, 

Thank you for submitting your manuscript entitled "Single-cell direct isoform sequencing reveals isoform switch and locus specific transposable elements regulation during preimplantation embryo development" for consideration as a Research Article by PLOS Biology. Please accept my sincere apologies for the delay in getting back to you as we consulted with an academic editor about your submission.

Your manuscript has now been evaluated by the PLOS Biology editorial staff, as well as by an academic editor with relevant expertise, and I am writing to let you know that we would like to send your submission out for external peer review.

Once your full submission is complete, your paper will undergo a series of checks in preparation for peer review. After your manuscript has passed the checks it will be sent out for review. To provide the metadata for your submission, please Login to Editorial Manager (https://www.editorialmanager.com/pbiology) within two working days, i.e. by Jun 27 2023 11:59PM.

Kind regards,

Richard

Richard Hodge, PhD

rhodge@plos.org

PLOS

---

## [Decision Letter · Decision Letter 1]

26 Jul 2023

Dear Dr Fan,

Thank you for your patience while your manuscript "Single-cell direct isoform sequencing reveals isoform switch and locus specific transposable elements regulation during preimplantation embryo development" was peer-reviewed at PLOS Biology. Please accept my apologies for the delays that you have experienced during the peer review process. Your manuscript has now been evaluated by the PLOS Biology editors, an Academic Editor with relevant expertise, and by three independent reviewers. 

In light of the reviews, which you will find at the end of this email, we would like to invite you to revise the work to thoroughly address the reviewers' reports.

As you will see, the reviewers are generally positive about the findings but raise overlapping concerns that the manuscript lacks some analyses to control for technical artefacts and to provide additional biological relevance for the findings. In addition, Reviewer #3 notes that additional functional insight into the roles of specific isoforms in driving embryo development should be provided, as well as lineage tracing data to investigate the functional role of transposable elements. After discussions with the Academic Editor, we agree with the reviewer that these insights would strengthen the manuscript and would be required to consider a revised version at the journal.

Given the extent of revision needed, we cannot make a decision about publication until we have seen the revised manuscript and your response to the reviewers' comments. Your revised manuscript is likely to be sent for further evaluation by all or a subset of the reviewers.

**IMPORTANT - SUBMITTING YOUR REVISION**

*Re-submission Checklist*

*Published Peer Review*

*PLOS Data Policy*

*Blot and Gel Data Policy*

Sincerely,

Richard

Richard Hodge, PhD

rhodge@plos.org

REVIEWS:

Reviewer #1: Wang, Shi et al present a study to observe full length transcripts using nanopore sequencing in murine preimplantation embryos on a single cell level. This technique aids the identification of transcript isoforms as well as transposable elements. The authors conclude on a switch in isoform diversity and dynamics, which coincides with zygotic genome activation (ZGA). Additionally, the technique allows for the observation of transposable elements (TEs) on a family as well as locus specific level. 

Although the authors present an interesting adaptation of the SMART-seq2 technique for low input samples, the manuscript seems to lack several key analyses and controls, as well as biological interpretation of their findings as described in more detail below. Additionally, several minor comments are listed below to improve the clarity of the text. 

Major comments

1. In line 182-184 the authors state that genes expressing more isoforms are among highly expressed genes. However, this could be caused by a sampling artifact considering that at this sequence depth it is more likely to detect more isoforms of genes that are highly transcribed compared to genes that are expressed at lower levels. The authors should confirm the lower isoform diversity of lowly expressed genes by comparing RT-PCR and sanger sequencing results of high and low expressed genes as was done for other findings in their study. 

2. In line 209-213, the authors discuss isoform candidates that are truncated at the 3' end. They state (as a biological finding) that there are no isoforms detected without poly (A) tail. This conclusion is in my view, incorrect, considering that the technique used (Smart-seq2) uses a poly-A-based RT approach and thus requires the transcript to have a poly(A) tail (as a consequence, SMART-seq-2 cannot naturally detect transcripts without poly(A) tails). 

3. Along these lines, the authors claim that isoforms are much more frequently truncated at the 5' end. Can the authors confirm that this is indeed a biological finding and not instead a result of incomplete reverse transcription? 

4. In figure 4B a DNA gel is shown accompanied by the statement that shorter isoforms are more abundant than the full-length transcript. Although this may very well be the case, a DNA gel can be hard to interpret for this purpose as longer fragments will appear brighter than shorter fragments due to the nucleotide staining. The authors should consider confirming this finding by bioanalyzer or by other more quantitative techniques. 

5. It is a novel and interesting finding that DNA transposons are found in this dataset to be expressed to a similar degree as LINEs, SINEs and LTRs. However, this finding does not get explained or highlighted by the authors. This should be added to the discussion.

6. According to the methods, for the TE analysis, only transcripts fully mapped to the TE sequence are taken into consideration. However, several studies (among which a recent study by Modzelewski, Cell 2021) have identified transcript isoforms where TEs are used as an alternative promoter for gene expression. It would be highly relevant for the authors to investigate chimeric transcripts containing both genic and TE regions in their dataset. 

Minor comments

1. The text should be thoroughly checked for spelling and grammar mistakes for clarity and correctness. Several examples of spelling errors (incomplete list); 

a. Line 74; "Calss1" should be "class I".

b. Line 85; "Lous" should be "locus".

c. Figure 5b + legend "Expressed TE locus" should be "Expressed TE loci".

d. In the methods in line 419 the authors state that expression values are normalized by copy number per 100,000 mapped reads. However, CPM is typically per 1,000,000 mapped reads. 

2. Line 77 introduces MERVL and MT2_Mm. It should be made clear the MT2_Mm is the LTR of the internal (originally coding) sequence of MERVL for a better understanding of the author's statement. 

3. On several occasions the text is currently missing references. For example;

a. Line 82-85; there are several reviews that discuss this problem and computational tools/wet lab techniques that address this problem that should be cited here. 

b. Line 132 after the statement that the mouse oocyte/zygote contains more RNA molecules compared to later stages. 

c. Line 181 a reference is missing for this public (?) SCAN-seq dataset and method. 

d. Line 286; there are several reviews that have highlighted this particular challenge of TE biology that can be cited (e.g. Lanciano and Cristofari, Nature Reviews Genetics 2020) 

4. All figure legends and labels should be improved for clarity. In particular; 

a. Figure 2 lacks information on how the rows in heatmap D-E are ordered within each cluster. 

b. Figure 5a lacks information on what the y-axis reflects (actual raw counts? Normalized counts? Number of TEs that were identified or number of counts that are inside TEs?). 

c. Figure 5C legends also does not contain information on whether the dots represent individual blastomeres or TE loci. 

d. The legend of Figure 5D states that unique loci are mapped (n=3894), but this number would make more sense if the heatmap shows unique TE families (e.g. MT2_Mm). Is this correct? 

e. Similar to figure 2, it is unclear how the rows of the heatmaps in figure 5 (5D, 5G and 5H) are ordered/clustered. This should be mentioned in the figure legend. Additionally, it should be highlighted in the legend that Figure 5G-H only show examples of individual loci and does not reflect all insertions of a specific family. 

5. Line 303-304; The statement "Our data shows that Lx7 …" should either be toned down or moved to the discussion as this is a very strong statement without experimental proof for support. 

6. Line 352; "On the other hand, abnormal expression of TE is harmful to human"; it should be specified that this is the case for most differentiated tissues, not during embryo development. 

Reviewer #2 (Jong Kyoung Kim, signs review): Summary: In this manuscript, the authors developed a protocol for long-read single-cell RNA sequencing to quantify the expression levels of isoforms and locus-specific transposable elements in single cells. The protocol, modified from SCAN-seq (using nanopore sequencing) to use PacBio HiFi sequencing, was applied to mouse oocytes, zygotes and blastomeres. They characterized the dynamic isoform switching patterns and locus-specific expression of transposable elements (TEs) during mouse embryonic development. 

The present manuscript is potentially important because it provides a valuable single-cell resource for understanding the complete landscape of gene expression during early embryonic development by profiling individual isoforms and locus-specific TEs. However, there are several major points that should be addressed to substantiate the authors' claims.

Major points:

1. The authors used SIRV spike-ins to evaluate the accuracy of quantifying isoform expression levels. For the SIRV spike-ins, they should cite the bioRxiv preprint by Paul et al., (https://doi.org/10.1101/080747). Fig. 1D does not provide any information on the accuracy of measuring isoform expression levels. The authors should provide this information by comparing the true and estimated isoform expression levels per cell, and by showing the relationship between technical noise (squared CV) and the mean isoform expression levels. I'm also wondering if the accuracy and technical noise of quantifying isoform expression in single cells are affected by the total amount of mRNAs in the cells. This can be done by evaluating the accuracy and technical noise of isoform quantification for each developmental stage, since the total amount of mRNAs per cell would decrease from oocytes to blastomeres.

2. Fig. S1A and B: Since the number of isoform types is strongly influenced by the expression levels of genes (Fig. S1C and D), the rich isoform diversity observed in early mouse embryos may be an artefact by their increased total amount of mRNAs per cell. The authors should show that technical replicates of a cell from early embryos with reduced total amount of mRNAs have similar isoform diversity. The technical replicates of a cell can be obtained by splitting a cell lysate into multiple aliquots. The same also applies to Fig. 3C. 

3. A list of genes showing dynamic isoform switching during early embryonic development should be provided as a supplementary table, and their shared biological properties should be systematically characterized and discussed. 

4. Fig. 5G and H: I'm wondering if neighboring genes of TE copies with similar expression patterns show more correlated expression during embryonic development. The role of TEs as putative regulators should be further characterized. 

5. The discussion section seems truncated and abruptly ended. The limitations of the proposed protocol and the current study should be discussed. 

Minor comments:

1. The terms "3' prime" and "5' prime" through the manuscript and figures should be replaced with "3'" (or "3-prime") and "5'" (or "5-prime"), respectively. 

2. Many typos in the manuscript (e.g. "down regulated" in line 156, "showes" in line 241, and "is consists of" in line 305)

3. Line 278-279: The sentence is not matched with Fig. 5C.

Reviewer #3: In this manuscript, the authors present a novel approach utilizing HIT-scISOseq, a hifi long-read sequencing technique from PacBio, to investigate isoform switching during embryo development in mouse preimplantation embryos. The authors have previously published a method called SCAN-seq (Fan X et al. PLoS Biol. 2020) for long-read single-cell RNA sequencing in mouse preimplantation embryos. In this study, they aim to enhance transcript abundance estimation by incorporating unique molecular identifiers (UMIs) and spike-in correction and evaluate the capability of the modified platform to identify isoforms.

The sequencing component of this study introduces two noteworthy advancements. Firstly, the authors successfully adapt the HIT-scISOseq method for low-throughput applications, enabling the quantification and identification of isoforms associated with not only alternative splicing but transposable elements. Secondly, they classify transcripts into various types, such as 3' prime partial, 5' prime partial, complete, and insertion, and perform quantitative analyses to assess changes in these categories.

However, it remains challenging to discern the specific benefits of this approach in revealing or supporting biological themes related to embryo development. Consequently, I recommend some revisions to enhance the impact and citation potential of this paper.

Major points

1. The number of blastomeres used in this study appears to be limited, with only 161 cells analyzed from various stages of preimplantation embryos. This sample size may not provide sufficient statistical power to discern cellular differences within each stage. Additionally, the rationale behind utilizing single-cell RNA sequencing instead of bulk-long read sequencing technology is not clearly explained. It would be beneficial to clarify the necessity of a single-cell-based approach in this study. In particular, the selection of only 13 blastocyst cells raises concerns about their representativeness, as these cells appear to be more clustered compared to cells from other stages (as shown in Figure 2B and 2C).

2. Isoforms can play distinct roles based on various factors such as changes in expression levels, structural conformation, loss of enzymatic activity, and rewiring of protein interactions. To provide a more comprehensive understanding of isoform dynamics and their implications in reprogramming during development, the authors should conduct further analyses. One approach could involve leveraging protein residue-level function annotation tools, such as FunFam from Christine A. Orengo, to identify functional dissimilarities among isoforms. By employing these tools, the authors may gain insights into the specific isoforms driving the developmental stages of preimplantation embryos. Integrating such analyses into the study would enhance our understanding of the functional implications of isoform switching during embryo development.

3. An important consideration is that even when using different embryos, the presence of inserted transposable elements (TEs) should be preserved if these TEs play a crucial role in development. Therefore, the identification of functional TEs can offer an opportunity for lineage tracing across different stages of development. This aspect underscores the significance of employing a single-cell-based approach. By characterizing the expression and activity of TEs at the single-cell level, the authors can potentially unravel their functional roles and trace the lineage of cells across different developmental stages. This approach would provide valuable insights into the impact of TEs on developmental processes, further justifying the use of a single-cell-based approach in this study.

Minor points

- Please show loadings when you make PCA plots in figure 2B and 2C. 

- Please add more information for figure captions and legends (e.g. -1 to 1 in 2D). 

- Typo: Page 10 line 74 calss1 -> class1

---

## [Decision Letter · Decision Letter 2]

7 Dec 2023

Dear Dr Fan,

Thank you for your patience while we considered your revised manuscript "Single-cell direct isoform sequencing reveals isoform switch and locus specific transposable elements regulation during preimplantation embryo development" for publication as a Research Article at PLOS Biology. Please accept my apologies for the delays that you have experienced during this round of the peer review process. This revised version of your manuscript has been evaluated by the PLOS Biology editors, the Academic Editor and the original reviewers.

Based on the reviews, I am pleased to say that we are likely to accept this manuscript for publication, provided you satisfactorily address the remaining points raised by Reviewer #1. In addition, we would strongly encourage you to enlist the services of a professional editing service or a native English-speaking colleague to improve the overall quality of the writing in the manuscript at this stage. 

Please also make sure to address the following data and other policy-related requests that I have provided below (A-I):

(A)After discussions with the editorial team, we think your manuscript would be a better fit as a Resource Article. During resubmission, we would be grateful if you could please tick ‘Methods and Resources’ as the article type in the drop down menu.

(B)We would like to suggest the following modification to the title, to make it more compelling and accessible for our broad readership: 

“Single-cell analysis of isoform switching and transposable element expression during preimplantation embryonic development”

(C) In addition, we would also like to suggest the following edits to the Abstract to improve the quality of the writing. In addition, we note that the Abstract is lacking a conclusion sentence that we think should be added at this stage. The conclusion sentence would highlight the relevance of the work and/or describe what future work the dataset now enables for instance. 

“Alternative splicing is an essential regulatory mechanism for development and pathogenesis. Through alternative splicing one gene can encode multiple isoforms and be translated into proteins with different functions. Therefore, this diversity is an important dimension to understand the molecular mechanisms governing embryo development. Isoform expression in preimplantation embryos has been extensively investigated, leading to the discovery of new isoforms. However, the dynamics of isoform switching of different types of transcripts throughout development remains unexplored. Here, using single-cell direct isoform sequencing in over one hundred single blastomeres from the mouse oocyte to blastocyst stage, we quantified isoform expression and found that 3-prime partial transcripts lacking stop codons are highly accumulated in oocytes and zygotes. These transcripts are not transcription by-products and might play a role in the maternal to zygote transition (MZT) process. Long-read sequencing also enabled us to determine the expression of transposable elements (TEs) at specific loci. In this way, we identified 3894 TE loci that exhibited dynamic changes during preimplantation development, likely regulating the expression of adjacent genes.”

(D) You may be aware of the PLOS Data Policy, which requires that all data be made available without restriction: http://journals.plos.org/plosbiology/s/data-availability. For more information, please also see this editorial: http://dx.doi.org/10.1371/journal.pbio.1001797

-Supplementary files (e.g., excel). Please ensure that all data files are uploaded as 'Supporting Information' and are invariably referred to (in the manuscript, figure legends, and the Description field when uploading your files) using the following format verbatim: S1 Data, S2 Data, etc. Multiple panels of a single or even several figures can be included as multiple sheets in one excel file that is saved using exactly the following convention: S1_Data.xlsx (using an underscore).

-Deposition in a publicly available repository. Please also provide the accession code or a reviewer link so that we may view your data before publication. 

Figure 1B-D, 2A-E, 2G-I, 3B-C, 4D-E, 5A-D, 5G-H, S2A-E, S2G, S3A-E, S4A-B, S5A-D

(E) We ask that you please deposit the scRNA-seq dataset in a public data repository, such as the GEO (https://www.ncbi.nlm.nih.gov/geo/). Please provide the accession number/URL of the deposition in the Data Availability Statement and ensure that the deposition is made publicly available at this stage. 

(F) Please also ensure that each of the relevant figure legends in your manuscript include information on *WHERE THE UNDERLYING DATA CAN BE FOUND*, and ensure your supplemental data file/s has a legend.

(G) We require the original, uncropped and minimally adjusted images supporting all blot and gel results reported in the following Figures:

Figure 3E, 4B, S3F

We will require these files before a manuscript can be accepted so please prepare and upload them now. Please carefully read our guidelines for how to prepare and upload this data: https://journals.plos.org/plosbiology/s/figures#loc-blot-and-gel-reporting-requirements

(H) Thank you already providing the underlying code for the HIT-scISO-seq analysis pipeline in Github (https://github.com/shizhuoxing/scISA-Tools). However, we ask that you please attach this deposition to the Zenodo repository (https://zenodo.org/) to ensure long-term maintenance of the deposition and that it is given a DOI. 

(I) Please ensure that your Data Statement in the submission system accurately describes where your data can be found and is in final format, as it will be published as written there. 

We expect to receive your revised manuscript within two weeks. 

*Published Peer Review History*

*Press*

Kind regards,

Richard

Richard Hodge, PhD

rhodge@plos.org

Reviewer remarks:

Reviewer #1: The authors have added quite some more analyses to TEs (as myself and the other reviewers had suggested), in particular the neighboring gene analysis. 

Overall, they addressed the comments I raised fully. However, I again strongly suggest the authors and the editorial team to improve the text for grammar correctness and clarity. Additionally, one minor suggestion is that I think the authors made a mistake in the label of figure 5C. As far as I understand the figure shows TE expression on a locus specific level in the upper panel and on a subfamily level on the lower level, however the legend states the opposite. 

Reviewer #2 (Jong Kyoung Kim, signs review): The authors have satisfactorily addressed my comments.

---

## [Editor Report · Decision Letter 3]

18 Jan 2024

Dear Dr Fan,

Thank you for the submission of your revised Methods and Resources entitled "Single-cell analysis of isoform switching and transposable element expression during preimplantation embryonic development" for publication in PLOS Biology. On behalf of my colleagues and the Academic Editor, Bon-Kyoung Koo, I am delighted to let you know that we can in principle accept your manuscript for publication, provided you address any remaining formatting and reporting issues. These will be detailed in an email you should receive within 2-3 business days from our colleagues in the journal operations team; no action is required from you until then. Please note that we will not be able to formally accept your manuscript and schedule it for publication until you have completed any requested changes.

PRESS

Sincerely, 

Ines

--

Ines Alvarez-Garcia, PhD

Senior Editor

PLOS Biology

on behalf of

Richard Hodge, PhD

Senior Editor

PLOS Biology

rhodge@plos.org